# Bacterial Number and Genetic Diversity in a Permafrost Peatland (Western Siberia): Testing a Link with Organic Matter Quality and Elementary Composition of a Peat Soil Profile

**Andrey S. Aksenov** [1,*], **Liudmila S. Shirokova** [2,3], **Oksana Ya. Kisil** [1], **Sofiia N. Kolesova** [1], **Artem G. Lim** [4], **Darya Kuzmina** [4], **Sophie Pouillé** [5], **Marie A. Alexis** [5], **Maryse Castrec-Rouelle** [5], **Sergey V. Loiko** [4] and **Oleg S. Pokrovsky** [3]

1  Arctic Biomonitoring Laboratory, Northern (Arctic) Federal University, 17 Naberezhnaya Sev. Dviny, 163002 Arkhangelsk, Russia; kisil.oksana.jroslavovna@gmail.com (O.Y.K.); sofiavanwyn@gmail.com (S.N.K.)
2  N. Laverov Federal Center for Integrated Arctic Research, Russian Academy of Sciences, 23 Naberezhnaya Sev. Dviny, 163000 Arkhangelsk, Russia; liudmila.shirokova@get.omp.eu
3  Geosciences and Environment Toulouse, UMR 5563 CNRS, 14 Avenue Edouard Belin, 31400 Toulouse, France; oleg.pokrovsky@get.omp.eu
4  BIO-GEO-CLIM Laboratory, Tomsk State University, 36 Lenina, Build. 1, 634050 Tomsk, Russia; lim_artyom@mail.ru (A.G.L.); kuzmina.d.m.95@gmail.com (D.K.); s.loyko@yandex.ru (S.V.L.)
5  UMR 7619 Métis, Sorbonne Université, F-75252 Paris, France; 24sophiepouille@gmail.com (S.P.); marie.alexis@sorbonne-universite.fr (M.A.A.); maryse.rouelle@sorbonne-universite.fr (M.C.-R.)
*  Correspondence: a.s.aksenov@narfu.ru

**Abstract:** Permafrost peatlands, containing a sizable amount of soil organic carbon (OC), play a pivotal role in soil (peat) OC transformation into soluble and volatile forms and greatly contribute to overall natural $CO_2$ and $CH_4$ emissions to the atmosphere under ongoing permafrost thaw and soil OC degradation. Peat microorganisms are largely responsible for the processing of this OC, yet coupled studies of chemical and bacterial parameters in permafrost peatlands are rather limited and geographically biased. Towards testing the possible impact of peat and peat pore water chemical composition on microbial population and diversity, here we present results of a preliminary study of the western Siberia permafrost peatland discontinuous permafrost zone. The quantitative evaluation of microorganisms and determination of microbial diversity along a 100 cm thick peat soil column, which included thawed and frozen peat and bottom mineral horizon, was performed by RT-PCR and 16S rRNA gene-based metagenomic analysis, respectively. Bacteria (mainly *Proteobacteria*, *Acidobacteria*, *Actinobacteria*) strongly dominated the microbial diversity (99% sequences), with a negligible proportion of archaea (0.3–0.5%). There was a systematic evolution of main taxa according to depth, with a maximum of 65% (*Acidobacteria*) encountered in the active layer, or permafrost boundary (50–60 cm). We also measured C, N, nutrients and ~50 major and trace elements in peat (19 samples) as well as its pore water and dispersed ice (10 samples), sampled over the same core, and we analyzed organic matter quality in six organic and one mineral horizon of this core. Using multiparametric statistics (PCA), we tested the links between the total microbial number and 16S rRNA diversity and chemical composition of both the solid and fluid phase harboring the microorganisms. Under climate warming and permafrost thaw, one can expect a downward movement of the layer of maximal genetic diversity following the active layer thickening. Given a one to two orders of magnitude higher microbial number in the upper (thawed) layers compared to bottom (frozen) layers, an additional 50 cm of peat thawing in western Siberia may sizably increase the total microbial population and biodiversity of active cells.

**Keywords:** permafrost; peat; pore water; bacteria; number; metabolism; active layer

## 1. Introduction

In contrast to the extensive research centered on bacterial abundance, diversity and metabolic activity in permanently and seasonally frozen mineral soils from high-latitude Canadian High Arctic [1–3], Spitsbergen [4], Siberia [5–7], Alaskan Arctic [8,9], Greenland [10] and Antarctica [11–13], frozen peat (organic) environments remain poorly characterized in terms of the physiological diversity and the metabolic potential of bacteria. Permafrost peatlands represent the largest reservoir of currently frozen soil organic carbon, susceptible to microbial degradation under ongoing permafrost thaw and the increase of active layer thickness (ALT). For example, the western Siberia Lowland (WSL) exhibits a typical peat thickness of 1 to 3 m [14] and a maximal active layer depth of 30 to 80 cm [15–19]. Here, a significant reservoir of organic carbon and nitrogen is currently frozen but may become available for microbial processing in the next century. However, the microbial metabolic activity and even the number of bacteria in the thawed versus frozen parts of the peat soil cores are only beginning to be explored ([20] and references therein).

Over the past decade, several studies (i.e., [1,3,21–29]) have addressed microbial communities along a soil column that includes both seasonally thawed and permanently frozen horizons; see [30] for a review. However, these studies dealt with samples that contained a relatively small amount of organic carbon (<5–10% of $C_{org}$) and therefore characterized the bacterial activity and diversity in the mineral rather than the organic frozen layer. Much less is known of the microbial diversity in peatland soils subjected to permafrost impact ([31–33]). Moreover, some studies have documented significant differences in the microbial community structure between different soil horizons, including in the organic topsoil of western Siberia [34] and the subarctic soil of Alaska [23,35] and the Canadian High Arctic [1]. A few studies have used a metagenomic approach to address the diversity of microbial communities in arctic peat [31,33,36,37] and mineral [38–41] soils linked to the methane cycle [42,43], and several authors have characterized microbial biogeography in surface waters of permafrost regions [44–47].

However, despite this extensive work on microbial diversity in permafrost environments, the main external (physico-chemical) parameters governing microbial populations remain poorly understood. In this work, we aimed at testing the functional link between microbial diversity along various layers of a peat core and the organic and inorganic chemical composition of peat along the same core. The novelty of the present study consists of relating the genetic diversity of microorganisms to the organic matter quality of the peat core and elementary composition of both solid and liquid compartments of the core sampled during the period of maximal active layer thickness; this includes both the thawed layer and frozen organic horizons, down to the mineral horizon.

In agreement with previous studies [3,4,48–52], we hypothesized a decrease in both the number and genetic diversity of bacteria with the increase in soil depth, with an abrupt drop below the permafrost boundary. We further hypothesized, based on a previous assessment of metabolic diversity of microorganisms along the peat core of the WSL [20], that there will be a functional control, due to the organic chemical composition and trace metal concentration, of the relative abundance of dominant taxa over the surface and deep horizons, with a possible maximum of specific groups at the boundary of the active-permafrost layer. To test these hypotheses, the specific goals of this work are as follows: (i) to quantify the abundance and genetic diversity of bacteria and archaea over the full depth of the peat core, including the underlying mineral horizon and (ii) to identify potential organic substrates and trace elements that may be controlling microbial number and diversity in both the frozen and unfrozen parts of the peat core. Achieving these goals should allow us to predict the possible evolution of the soil microbial community due to downward migration of the active layer boundary, including its reaching of underlying mineral horizons due to ongoing permafrost thaw in western Siberia. This should eventually allow upscaling of these results to other permafrost peatlands of northern Eurasia.

## 2. Study Site and Methods

### 2.1. Peat Core Collection and Temperature Pattern of the WSL Peat Profile

Peat core samples were collected on 5 August 2017 in the palsa peat bog at the watershed divide of the discontinuous permafrost zone near the Khanymey test site of the northern part of the West Siberian Lowland (Figure 1). A detailed description of the study site is given elsewhere [20]. The dominant soil catenae are flat-mound bogs along the watershed divides, which comprise the Hemic Cryic Histosols on mounds. There, the peat (1.0–1.5 m thick) has developed on sands with minor amounts of clay and silt. The mounds (50 ± 20% of total palsa area) and depressions are less than 1 m in diameter, and the depth of the depressions is 40 to 50 cm. The average thickness of the active layer is 41 ± 5 cm on the mounds.

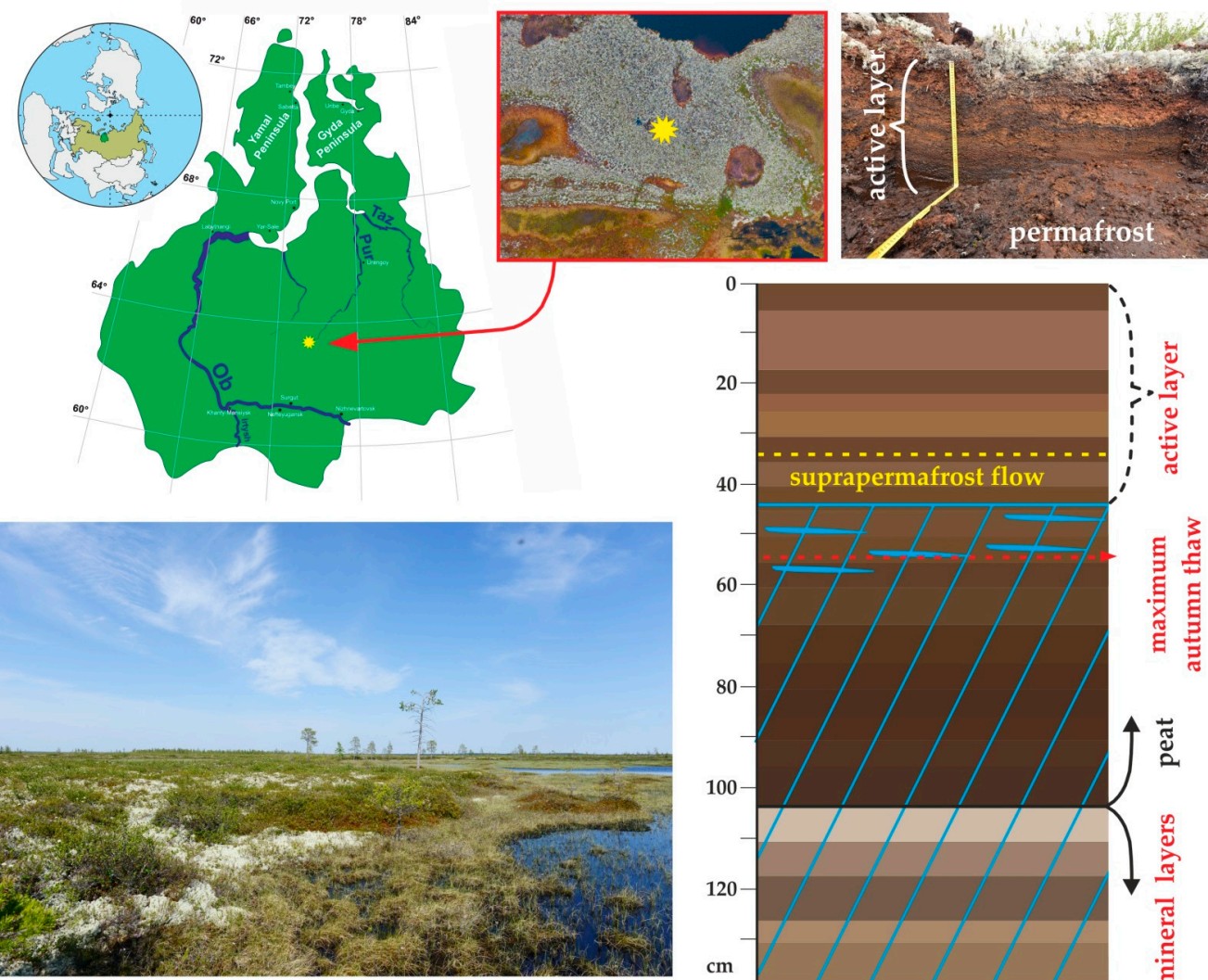

**Figure 1.** Ground view of the study site (63°47′50 N,10″, 75°33′21,90″ E), a photo of the soil pit and environmental context of the frozen palsa peat bog and a schematic profile of the peat column with relevant horizons. Photo made by Loiko S.V.

Peat core samples from mounds were extracted using a motorized Russian peat corer (UKB-12/25 I, Saint Petersburg, Russia) with a 4 cm diameter corer, sterilized with 40% ethanol prior to each extraction. Core subsections were taken each 5–10 cm over the length of the peat core. The samples were kept at −20 °C during transport and storage. Sterility of the part of the core collected for microbiological analysis was achieved using peat exclusively from the interior of the core (>1 cm from the core liner), following conventional procedures [20,53]. At every 5–10 cm of depth, soil samples were taken with a sharp,

sterile, single-use knife and placed in sterile PVC doubled-zipped bags. All subsequent subsampling was performed under sterile, protected conditions in the laminar hood box (A100).

To collect peat ice, the frozen peat core was cut into 10 cm layers using a ceramic blade and thawed at room temperature. Immediately after full thawing of the frozen core, peat water was extracted through 100 μm pre-cleaned nylon net by applying a local confining pressure (1–5 kg cm$^{-2}$, depending on sample humidity). The obtained slurry was centrifuged for 15 min at 3500 rpm, and the supernatant was filtered through a 0.45 μm acetate cellulose filter, with the first 10 mL of filtrate discarded. The water from the thawed part of the peat core, above the permafrost table, was extracted using the same method but in the field and immediately after peat sampling.

The degree of peat decomposition was determined as described previously [20]. The peat is essentially composed of *Sphagnum rubellum*, lichens (*Cladonia stellaris*), dwarf shrubs, green mosses, pinewood, and pine and birch bark, as described in detail elsewhere [20]. The peat color was measured in color coordinates CIE lab using a Rite VS450 (X-Rite, Inc., Grand Rapids, MI, USA).

The radiocarbon age of peat was determined by the liquid scintillation method using a Quantulus 1220 radiometer (Wallac, Turku, Finland) in the laboratory of the Tomsk Collective Use Center (Russian Academy of Sciences). Calibration of the radiocarbon age in the calendar was conducted using the OxCal online program and the IntCal 20 calibration curve [54].

The annual pattern of temperature at different depths was recorded each 10 cm over a 0–1 m depth via in situ temperature sensors CAM-2 (IMCES SB RAS, Tomsk, Russia), installed 2 years before sampling of the peat core.

### 2.2. Total Chemical Analysis

The peat core was analyzed for concentrations of C, N, P, and K as well as major and trace elements. Nitrogen and carbon concentrations in dry peat samples were measured by Cu-O catalyzed dry combustion at 900 °C with ≤0.5% precision for standard substances (Thermo Flash 2000 CN Analyzer). Sulfur was measured by EMIA-Expert Carbon/Sulfur Analyzer (Horiba, Kyoto, Japan). For the measurement of the element concentrations, the samples were first processed in a clean room (class A 10,000). Approximately 100 mg of peat, previously ground with an agate mortar and pestle to a <1 mm fraction, was placed in Teflon (Savilex®, Eden Prairie, MN, USA) reactors with 6 mL bi-distilled $HNO_3$, 0.2 mL ultrapure HF and 1 mL ultrapure $H_2O_2$. Twelve reactors were loaded into a Mars 5 microwave digestion system (CEM, Paris, France) and exposed for 20 min at 150 °C. Each series of reactors was composed of 10 samples of peat, 1 certified lichen standard CRM 482 sample (from BCR, Brussels, Belgium) or other NIST soil standards (NIST 271a Montana and NIST 2709a San Joaquin) and 1 blank sample. The major and trace element concentrations were measured by inductively coupled plasma mass spectrometry (ICP-MS; Agilent 7500 ce) using a three-point calibration against a standard solution of known concentration. Indium and rhenium were used as the internal standards to correct for instrumental drift and eventual matrix effects. Further details of elementary analyses of peat are available in Stepanova et al. [55]. The elementary composition of peat was analyzed with a 5 cm step; the results from individual layers were averaged to match the 10 cm spatial resolution of microbiological analysis. In addition to total chemical analysis, labile P-PO$_4$ in aqueous leachates was measured using the colorimetric technique [56].

Filtered peat pore water from the active layer and peat ice were analyzed as described elsewhere [57,58]. In brief, filtered solutions were divided into two parts: (1) acidified (pH ~2) with ultrapure double-distilled $HNO_3$ for trace element (TE) analyses and (2) non-acidified for dissolved organic carbon (DOC), dissolved inorganic carbon (DIC) and anion analyses. DOC and DIC were analyzed using a Carbon Total Analyzer (Shimadzu TOC VSCN) with an uncertainty better than 3%. Major anion (Cl, $SO_4^{2-}$) concentrations were measured by ion chromatography (HPLC, Dionex ICS 2000), with an uncertainty of 2%.

Major cations (Ca, Mg, Na, K), Si and ~40 trace elements were determined with an Agilent ce 7500 ICP-MS, with In and Re as internal standards.

### 2.3. Organic Matter Isotopic Composition and Quality

In freeze-dried samples of different layers from the peat core, the $\delta^{13}C$ (expressed in ‰ of international PDB standard) and the $\delta^{15}N$ (expressed in ‰ of atmospheric $N_2$) were measured using an isotopic ratio mass spectrometer IRMS (Micromass Iso Prime) in the iEES, Paris, France. An analytical standard (tyrosine) was measured for each of the 4 samples. The analytical precision was 0.2 and 0.1‰ for the $\delta^{13}C$ and $\delta^{15}N$, respectively.

The organic matter of samples was characterized by Nuclear Magnetic Resonance (NMR) of solid-state $^{13}C$ using a Bruker Avance 500 NMR (Bruker Analytik GmbH, Rheinstetten, Germany) in IMPC (Sorbonne Université, Paris, France), with a resonance frequency of 125 MHz for $^{13}C$. A CP-MAS (cross-polarization magic angle spinning) was used, with a rotation speed of 14 KHz. The contact and relaxation time were 1 ms and 1 s, respectively. The chemical shift was quantified with respect to trimethilsilane (TMS) and adjusted with adamantane (1.85 ppm for $^1H$ and 38.52 ppm for $^{13}C$) as external references. Spectra deconvolution was performed with the Dmfit 2015 program [59]. The spectra were divided into 4 regions of chemical shift, as defined by [60]: alkyl C (0–45 ppm), O-alkyl C (45–110 ppm), aromatic C (110–160 ppm) and carboxylic C (160–210 ppm). The proportion of these functional groups, after deconvolution and integration, was obtained via adding the areas of peaks corresponding to the defined regions.

### 2.4. Analysis of Bacterial Number and Genetic Diversity

The extraction of genomic DNA from frozen peat samples was conducted at the Research Institute for Agricultural Microbiology (Pushkin, St.-Petersburg, Russia) using the Macherey-Nagel nucleic spin kit (Macherey-Nagel, Düren, Germany), following the manufacturer's instructions. Taxonomic analysis of the bacterial community was carried out using universal primers F515 (GTGCCAGCMGCCGCGGTAA) and R806 (GGAC-TACVSGGGTATCTAAT) on the variable region V3–V4 of the 16S rRNA gene, which is specific for a wide range of microorganisms, including bacteria and archaea. A rough estimate of the number of microorganisms (counting clones) in the samples was carried out using real-time PCR using the following primers: Eub338/Eub518 for bacteria and arc915f/arc1059r for archaea. As a control for bacteria, cloned fragments of the ribosomal operon *Escherichia coli* (Sigma) were used, and for the archaea, strain FG-07 *Halobacterium salinarum* (the strain was provided by G. Jurgens, University of Helsinki, Helsinki, Finland). The determination for each sample was carried out in triplicate. Quantitative estimates are given as to the number of 16S rRNA gene copies per gram of soil.

### 2.5. Statistical Treatment

Statistical treatment of the data included best-fit functions based on the Pearson correlation and one-way ANOVA with the STATISTICA version 8 software (StatSoft Inc., Tulsa, OK, USA). Linear regressions were used to examine the relationships among the total and culturable cell numbers, the total average well color development (AWCD) and the consumption of an individual group of substrates, and the chemical composition of the peat layer (C, N, major and trace elements). The criterion for a significant correlation between chemical and biological parameters was that the Pearson coefficients were higher than 0.5 ($R \geq 0.5$ and $p < 0.05$). The ANOVA method was used to test the differences in the cell number, elementary composition, organic matter quality parameters and shares of the 16S rRNA gene fragments of individual phylogenetic groups. Further statistical treatment of a complete set of a share of phylogenetic groups, inorganic (ICP-MS) and organic (NMR) chemical and isotopic composition and physical parameters of peat layer included principal components analysis (PCA). The PCA allowed testing of the effect of various parameters, the differences between peat layers in particular, on the chemical and physical parameters of peat and spatial distribution of microorganisms. Note that the selected peat core is highly

representative of the region of WSL permafrost peatlands. Multiple cores sampled within the same test site of Khanymey (discontinuous permafrost zone), from different mounds, demonstrated a high similarity of the chemical composition of both pore water and peat ice [58]. Due to the high homogeneity of the peat and underlaying mineral substrate of the region, we used a single soil core for representing the chemical diversity of solid and liquid phases and the metabolic activity of microorganisms ([20]).

## 3. Results

*3.1. Peat Formation History, Thermal Regime, Physical and Elementary Characteristics of the Peat Core and Chemical Composition of the Pore Water*

Accumulation of peat started $8878 \pm 74$ BP (depth of 100 cm). The age of peat at the ALT boundary (41 cm) is $1886 \pm 84$ BP, whereas at a depth of 19 cm, the peat is $374 \pm 89$ years old. This corresponds to the rate of peat accumulation of 0.084 mm $y^{-1}$ for the depth of 41 to 100 cm and 0.15 mm $y^{-1}$ for the 19–41 cm layer. The upper 19 cm of peat was growing with a rate of 0.51 mm $y^{-1}$. This trend in accumulation rate increasing with time reflects the dominance in the upper part of the core of sphagnum mosses, which exhibit better preservation due to their slower degradation rate.

The thermal regime of the peat mound allowed for the determination of the average thickness of the active layer as 45–50 cm (Figure 1). The degree of peat decomposition increased from the top to the bottom of the sampled column, from 5% in the upper 10–20 cm layer to 30% below 70 cm depth, whereas the bulk peat density increased from 20 kg $m^{-3}$ in the upper 0–2 cm layer to 200–250 kg $m^{-3}$ below 60 cm depth.

The physical parameters of peat as well as the major and trace element compositions of the peat and peat pore water are listed in Table 1 and Table S1 of the Supplementary Materials. The C and N concentrations in the peat core are illustrated in Figure 2. Non-systematic variations in the C and N contents were observed over the peat core length, with an abrupt decrease in C and N in the mineral layer. The $N_{tot}$ concentration exhibited a local maximum at 75 cm depth. The C:N ratio varied between 50 and 100 in the active layer, decreased to 40–45 in the 40–85 cm layer, then increased to 63–64 in the bottom layers.

**Table 1.** Physical and chemical parameters of the studied core. The temperature (T) of the core was measured via in situ sensors over the whole year (2018–2019). The isotopic and RMN data corresponded to slightly different depths, as listed in Table 2, and they were averaged to match the depth horizons of genomic analysis.

| Depth, cm | Substrate | Average Depth, cm | Average T, °C | Minimal T, °C | Maximal T, °C | Peat, Bulk Density, g/cm³ | Degree of Decomposition, % |
|---|---|---|---|---|---|---|---|
| 10–20 | peat | −15 | −0.047 | −8.4 | 11.3 | 0.16 | 5 |
| 30–40 | peat | −35 | −0.63 | −6.1 | 4.4 | 0.29 | 10 |
| 50–60 | peat | −55 | −0.8 | −4.1 | 0.2 | 0.29 | 17 |
| 70–80 | peat | −75 | −0.74 | −2.8 | −0.1 | 0.23 | 30 |
| 90–100 | peat | −95 | −0.65 | −2.3 | −0.1 | 0.23 | 32 |
| 100–110 | mineral | −105 | −0.58 | −1.3 | −0.1 | 1.30 | |
| Depth, cm | substrate | N, % | C, % | C:N | L* by CIE lab | a* by CIE lab | b* by CIE lab |
| 10–20 | peat | 0.79 | 44.6 | 56.1 | 43.7 | 9.9 | 16.5 |
| 30–40 | peat | 0.61 | 46.0 | 75.4 | 39.2 | 9.1 | 17.5 |
| 50–60 | peat | 0.83 | 40.6 | 48.9 | 33.2 | 8.7 | 17.5 |
| Depth, cm | substrate | N, % | C, % | C:N | L* by CIE lab | a* by CIE lab | b* by CIE lab |
| 70–80 | peat | 1.36 | 56.1 | 41.3 | 25.9 | 7.8 | 14.0 |
| 90–100 | peat | 0.84 | 54.41 | 64.8 | 25.3 | 6.9 | 12.3 |
| 100–110 | mineral | 0.03 | 1.905 | 63.5 | 66.2 | 3.9 | 10.6 |

**Table 1.** *Cont.*

| Depth, cm | Substrate | Average Depth, cm | Average T, °C | Minimal T, °C | Maximal T, °C | Peat, Bulk Density, g/cm³ | Degree of Decomposition, % |
|---|---|---|---|---|---|---|---|
| Depth, cm | substrate | S, % | P-lab, mg/kg | $\delta^{13}$C | $\delta^{15}$N | Alkyl C | O-alkyl C |
| 10–20 | peat | 0.38 | 0.76 | −24.7 | 4.4 | 24.05 | 71.9 |
| 30–40 | peat | 0.38 | 0.34 | −25.9 | 4.8 | 15.3 | 77.3 |
| 50–60 | peat | 0.29 | 0.06 | −26.4 | 4.25 | 26.2 | 63.9 |
| 70–80 | peat | 0.39 | 0.04 | −26.9 | 3.7 | 37.1 | 50.5 |
| 90–100 | peat | 0.22 | 0.03 | −26.2 | 4.1 | 47.1 | 39.3 |
| 100–110 | mineral | 0.05 | 0.00 | −26.9 | | | |

| Depth, cm | substrate | aromatic C | carboxylic C | Li peat | B peat | Na peat | Mg peat |
|---|---|---|---|---|---|---|---|
| | | | | | | ppb | |
| 10–20 | peat | 1.6 | 2.3 | 577 | 4518 | 372,422 | 296,970 |
| 30–40 | peat | 5.2 | 2.2 | 294 | 4544 | 103,287 | 327,233 |
| 50–60 | peat | 7.8 | 2.1 | 498 | 4434 | 72,538 | 194,873 |
| 70–80 | peat | 10.4 | 2.0 | 294 | 4892 | 29,929 | 169,643 |
| 90–100 | peat | 10.1 | 3.5 | 424 | 11,608 | 83,755 | 145,879 |
| 100–110 | mineral | | | 2441 | 3359 | 235,580 | 74,206 |

| Depth, cm | substrate | Al peat | P peat | K peat | Ca peat | Ti peat | V peat |
|---|---|---|---|---|---|---|---|
| | | | | | | ppb | |
| 10–20 | peat | 2,302,515 | 448,947 | 938,435 | 1,049,413 | 183,028 | 3059 |
| 30–40 | peat | 1,392,605 | 295,471 | 232,556 | 1,525,343 | 52,626 | 1199 |
| 50–60 | peat | 1,813,247 | 209,406 | 284,074 | 911,927 | 86,250 | 1987 |
| 70–80 | peat | 1,925,618 | 200,513 | 192,876 | 921,297 | 59,999 | 1936 |
| 90–100 | peat | 2,797,844 | 172,804 | 327,006 | 832,506 | 116,491 | 2942 |
| 100–110 | mineral | 1,748,772 | 22,357 | 991,020 | 279,474 | 218,869 | 1677 |

| Depth, cm | substrate | Cr peat | Mn peat | Fe peat | Co peat | Ni peat | Cu peat |
|---|---|---|---|---|---|---|---|
| | | | | | | ppb | |
| 10–20 | peat | 2504 | 10,234 | 1,311,464 | 281 | 1169 | 2138 |
| 30–40 | peat | 1966 | 5228 | 1,433,817 | 426 | 1083 | 594 |
| 50–60 | peat | 2745 | 5699 | 472,019 | 218 | 942 | 1152 |
| 70–80 | peat | 2793 | 2139 | 451,913 | 284 | 1841 | 2718 |
| 90–100 | peat | 3871 | 5225 | 519,426 | 522 | 1573 | 5064 |
| 100–110 | mineral | 1895 | 8516 | 290,073 | 177 | 263 | 914 |

| Depth, cm | substrate | Zn peat | As peat | Rb peat | Sr peat | Y peat | Zr peat |
|---|---|---|---|---|---|---|---|
| | | | | | | ppb | |
| 10–20 | peat | 15,946 | 1354 | 3086 | 9573 | 618 | 3614 |
| 30–40 | peat | 15,957 | 315 | 748 | 13,479 | 345 | 1356 |

| Depth, cm | substrate | Zn peat | As peat | Rb peat | Sr peat | Y peat | Zr peat |
|---|---|---|---|---|---|---|---|
| | | | | | | ppb | |
| 50–60 | peat | 7367 | 261 | 1185 | 8224 | 463 | 2777 |
| 70–80 | peat | 9325 | 391 | 694. | 8406 | 332 | 1843 |
| 90–100 | peat | 17,238 | 380 | 1084 | 8394 | 414 | 5941 |
| 100–110 | mineral | 2355 | 299 | 2854 | 7518 | 913 | 14,104 |

| Depth, cm | substrate | Nb peat | Mo peat | Cd peat | Sb peat | Cs peat | Ba peat |
|---|---|---|---|---|---|---|---|
| | | | | | | ppb | |
| 10–20 | peat | 476 | 155 | 130 | 260 | 189 | 27,243 |
| 30–40 | peat | 462 | 87 | 132 | 39 | 46 | 20,066 |
| 50–60 | peat | 863 | 88 | 71 | 29 | 72 | 22,801 |
| 70–80 | peat | 579 | 211 | 86 | 42 | 41 | 29,938 |
| 90–100 | peat | 887 | 108 | 72 | 37 | 65 | 39,452 |
| 100–110 | mineral | 924 | 48 | 14 | 139 | 83 | 69,278 |

**Table 1.** *Cont.*

| Depth, cm | Substrate | Average Depth, cm | Average T, °C | Minimal T, °C | Maximal T, °C | Peat, Bulk Density, g/cm³ | Degree of Decomposition, % |
|---|---|---|---|---|---|---|---|
| Depth, cm | substrate | La peat | Ce peat | Dy peat | Yb peat | Pb peat | U peat |
| | | | | ppb | | | |
| 10–20 | peat | 1184 | 2365 | 120 | 69 | 6485 | 80 |
| 30–40 | peat | 537 | 1167 | 72 | 32 | 1009 | 50 |
| 50–60 | peat | 775 | 1559 | 90 | 54 | 647 | 92 |
| 70–80 | peat | 534 | 1090 | 65 | 35 | 532 | 88 |
| 90–100 | peat | 727 | 1413 | 83 | 47 | 837 | 111 |
| 100–110 | mineral | 850 | 1498 | 108 | 86 | 1036 | 162 |
| Depth, cm | substrate | Humidity, % | | | | | |
| 10–20 | peat | 317 | | | | | |
| 30–40 | peat | 337 | | | | | |
| 50–60 | peat | 700 | | | | | |
| 70–80 | peat | 458 | | | | | |
| 90–100 | peat | 650 | | | | | |
| 100–110 | mineral | 82 | | | | | |

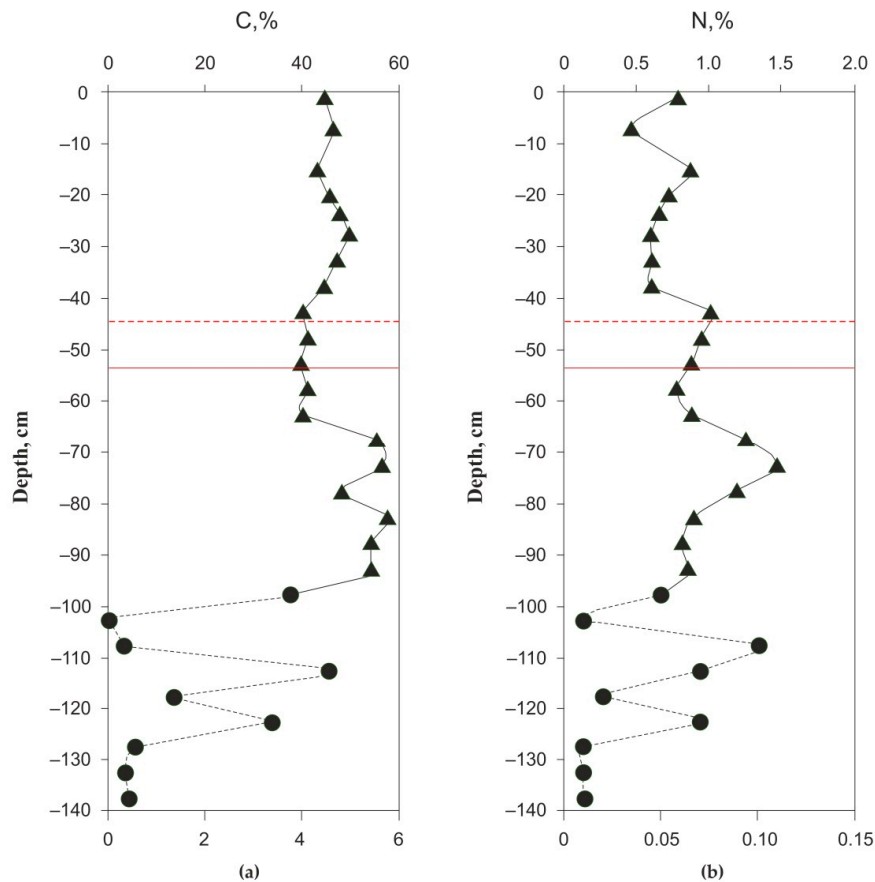

**Figure 2.** Total $C_{org}$ (**a**) and N (**b**) in the peat core as a function of depth. The actual depth of the active layer on the day of sampling is 43 cm, shown by dashed line, and the mean maximal depth at the end of the active season is ~54 cm, shown by solid line. The concentrations of C and N in the peat and mineral layer are represented by triangles (upper X axis) and circles (bottom X axis), respectively.

The upper 20 cm layer of peat was enriched in macro- (P, K, Mg) and micro-nutrients (Mn, Zn, Rb) compared to the rest of the peat core, without systematic variations in the depth below 20 cm (Table 1). Some trace elements of atmospheric origin (As, Cd, Sb, Cs, Tl, Pb) also enriched the upper 10–20 cm of the peat core. The other major and trace elements

varied non-systematically across the sampled depth and did not demonstrate a significant (at $p < 0.05$) difference between the active and the frozen layer.

The peat pore water from the active layer and dispersed ice from the permafrost horizons exhibited a vertical pattern of elementary composition that was different from that of the solid phase (Figure 3). The phosphorus demonstrated a maximum at 70–80 cm, whereas the majority of other elements exhibited a downward increase in concentration, with some of them, such as micro-nutrients (B, K, Mn, Ni, Zn, Mo, Ba) and atmospherically-deposited toxicants (As, Sb, Pb), also showing enrichment in the shallowest surface (0–10 cm) layer (Table S2).

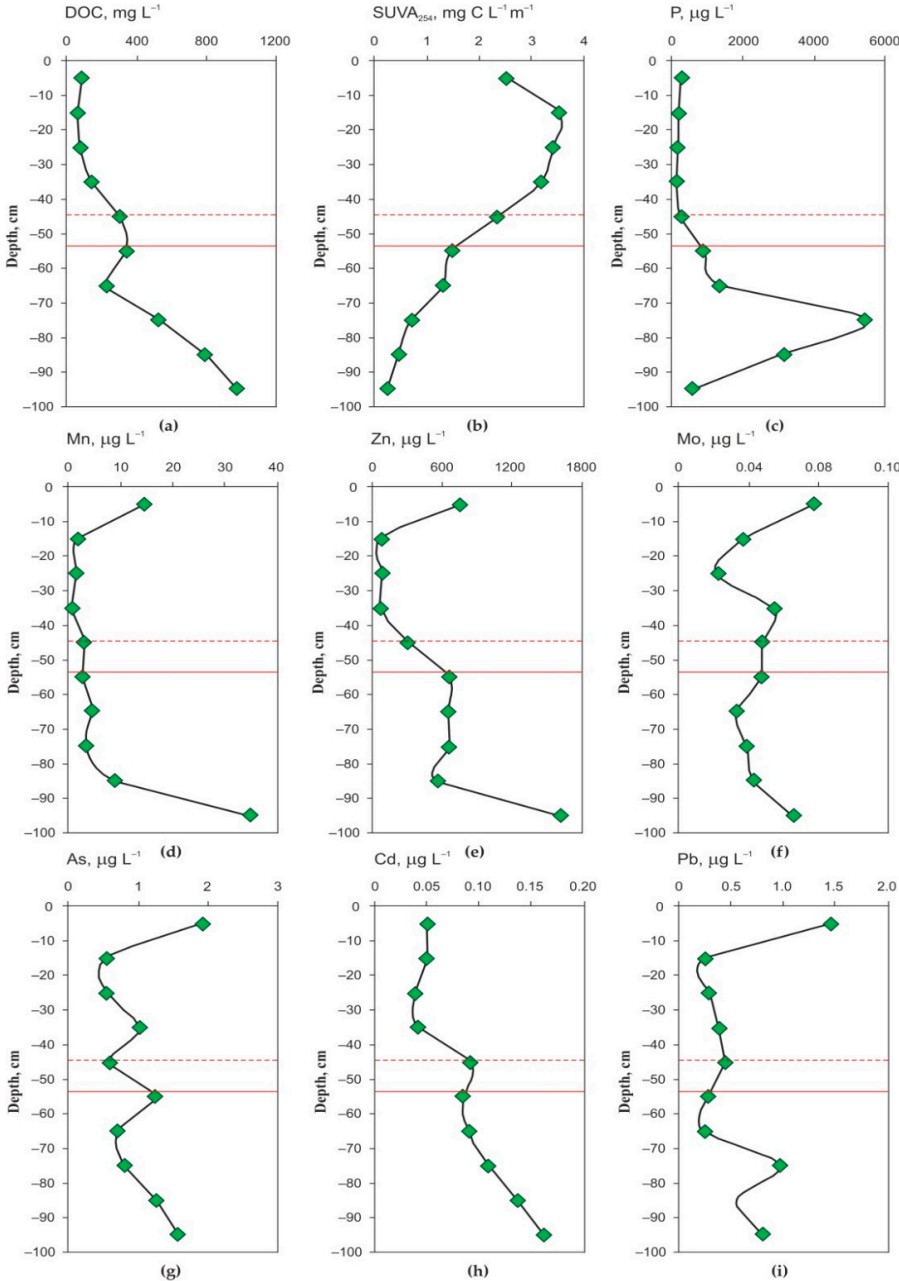

**Figure 3.** Concentrations of DOC (**a**), specific UV absorbance at 254 nm ($SUVA_{254}$) (**b**), P (**c**), Mn (**d**), Zn (**e**), Mo (**f**), As (**g**), Cd (**h**), and Pb (**i**) in the fluid phase of the peat core (pore water of the active layer and the permafrost ice). The actual depth of the active layer on the day of sampling is 43 cm, shown by dashed line, and the mean maximal depth at the end of the active season is ~54 cm, shown by solid line.

### 3.2. Organic Matter Isotopic Composition and Functional Group Analysis by NMR

The isotopic and organic matter (OM) quality characteristics are listed in Table 2. The $\delta^{13}C$ decreased significantly with depth ($p = 0.03$), showing a maximum at the 0–5 cm layer (−23.6‰) and exhibiting non-systematic variations between −26 and −27‰ below 60 cm depth. The $\delta^{15}N$ varied between +3 and +6‰, without any tendency with respect to depth. Based on Cross Polarization Magic-Angle Spinning (CP-MAS) solid-state $^{13}C$ NMR spectra, the ratio alkyl C:O-alkyl C and the proportion of aromatic C significantly increased with sample depth ($p = 0.05$ and 0.049, respectively). The proportion of carboxylic C (band at 170–175 ppm) over the sampled horizons is highly variable (between 1.2 and 3.5%) and did not show any trend with depth.

**Table 2.** Isotopic and NMR results of the CKH18-1T peat core. n.m. * stands for not measured.

| Depth | $\delta^{13}C$ | $\delta^{15}N$ | Alkyl C | O-Alkyl C | Aromatic C | Carbonyl C | Alkyl C: O-Alkyl C |
|---|---|---|---|---|---|---|---|
| | ‰ | ‰ | % | | | | |
| (0–5) | −23.6 | 2.8 | 16.7 | 79.8 | 0.0 | 3.5 | 0.21 |
| (19–21) | −25.8 | 6.0 | 31.4 | 64.1 | 3.3 | 1.2 | 0.49 |
| (30–35) | −25.9 | 4.8 | 15.3 | 77.3 | 5.2 | 2.2 | 0.20 |
| (65–75) | −26.7 | 3.3 | 30.5 | 57.5 | 10.4 | 1.7 | 0.53 |
| (75–85) | −27.1 | 4.1 | 43.7 | 43.5 | 10.4 | 2.4 | 1.00 |
| (85–95) | −26.2 | 4.1 | 47.1 | 39.3 | 10.1 | 3.5 | 1.20 |
| (105–115) | −26.9 | n.m * | n.m * | n.m * | n.m * | n.m * | n.m * |

### 3.3. Cell Number and Microbial Diversity as a Function of Depth

The total number of microorganisms according to the real-time PCR, integrated over 0–110 cm depth, amounted to $2.26 \times 10^{10}$ (Table S2). Most of the peat microorganisms were present in the active layer, where environmental conditions are relatively milder than in the permafrost. The number of microorganisms decreased almost 1000-fold with depth, from $1.35 \times 10^{10}$ in the upper 0–20 cm layer to $2.27 \times 10^{7}$ in the mineral 100–110 cm layer.

Analysis of the composition of microbial communities by 16S rRNA sequencing demonstrated that the members of domain bacteria predominated in the soil profile (Figure 4). Based on the 16S rRNA gene fragments, the share of individual phylogenetic groups varied with soil depth (Table 3). The members of *Proteobacteria* (38.5%) and *Acidobacteria* (30.6%) predominated the sequence. The share of *Firmicutes* (3.25%) and *Actinobacteria* (14.1%) members was relatively high. Overall, the diversity of bacterial phyla gradually decreased with depth (Figure 4a). However, at the depth of 90–100 cm, the diversity sharply increased, but then it decreased again in the mineral layer (100–110 cm). The percentage of *Acidobacteria* and *Actinobacteria* increased until the depth of 50–60 cm, then decreased, and increased again at the depth of 90–100 cm (*Actinobacteria*) or in the mineral layer (*Acidobacteria*). The share of *Proteobacteria* decreased with depth, from 26.4% (10–20 cm) to 10.8% (50–60 cm). From the depth of 70–80 cm, the proportion of *Proteobacteria* began to rise again, up to 89.2% at the depth of 100–110 cm. The phylum *Verrucomicrobia* was present in significant quantities at two non-adjacent depths: 10.3% at 10–20 cm and 5.3% at 90–100 cm. The share of *Firmicutes* was insignificant up to a depth of 90 cm. In the deeper peat layers, the percentage of this phylum increased sharply to 15.9%, whereas in the underlaying mineral layer (100–110 cm), *Firmicutes* accounted for 2.5% of total phyla of microorganisms.

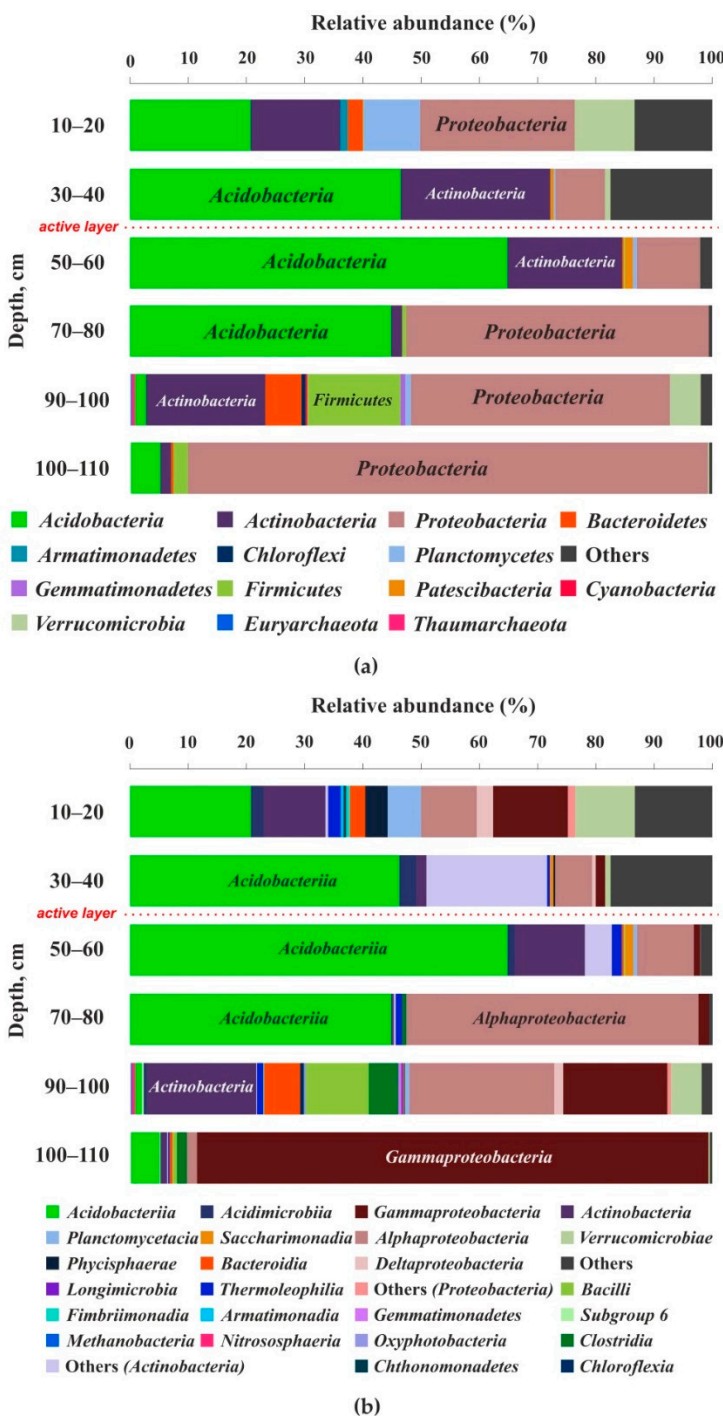

**Figure 4.** Phylum-level (**a**) and class-level (**b**) bacterial and archaeal community structure in the peat core based on 16S rRNA.

The phyla *Armatimonadetes* (0.2%), *Bacteroidetes* (1.6%), *Patescibacteria* (0.3%) and *Verrucomicrobia* (2.8%) were less diverse and were represented by one class each. The members of the class *Alphaproteobacteria* predominated and varied depending on the depth. The largest share of this class (50.2%) was related to the depth of 70–80 cm, and then it decreased (Figure 4b). Classes *Acidobacteriia* (30.3%) and *Gammaproteobacteria* (20.4%) were also represented by a large percentage. *Acidobacteriia* increased in number with depth from 20.5% at the surface to 64.7% in the 50–60 cm layer, and then decreased. However, in the mineral horizon (100–110 cm), *Acidobacteriia* were sizably higher (4.9%) than in the bottom peat layer (90–100 cm, 1.0%). The 16S rRNA gene sequences belonging to the

class *Gammaproteobacteria* were concentrated at the depth of 90–110 cm (17.7–87.5%). The class *Actinobacteria* exhibited sizable and non-systematic variations over depth, reaching the largest proportion (18.4%) at the depth of 90 to 100 cm. Classes *Bacteroidia* (6.1%), *Bacilli* (10.7%) and *Clostridia* (4.9%) were mostly present also at the same depth. Classes *Chthonomonadetes*, *Phycisphaerae*, *Planctomycetacia* and other non-identified bacteria predominated in the active layers (between 10 and 40 cm depth). Classes *Deltaproteobacteria* and *Verrucomicrobiae* were found in the greatest number at the surface and bottom horizon of the peat column: 2.9% and 10.3%, respectively, in the 10–20 cm layer; 1.6% and 5.3%, respectively, in the 90–100 cm layer. Class *Thermoleophilia* was almost uniformly distributed over the entire peat profile (Figure 4b). The classes *Gammaproteobacteria*, *Actinobacteria* and *Alphaproteobacteria* were most diverse and included six, five, and four orders, respectively. Classes *Acidobacteriia*, *Bacilli*, *Planctomycetacia* and *Verrucomicrobia* included two orders each. Other classes were represented only by one order.

**Table 3.** Relative proportion of class units of peat microorganisms over the peat column.

| Abbreviation | Taxon Unit (*Phylum: Class*) | Peat Horizon Depth, cm | | | | | |
|---|---|---|---|---|---|---|---|
| | | 10–20 | 30–40 | 50–60 | 70–80 | 90–100 | 100–110 |
| | | Share of 16S rRNA Gene Sequences in % of the Number of Obtained Sequences | | | | | |
| EuMeth | *Euryarchaeota; Methanobacteria* | 0.0 | 0.1 | 0.0 | 0.0 | 0.3 | 0.0 |
| ThNit | *Thaumarchaeota; Nitrososphaeria* | 0.0 | 0.0 | 0.0 | 0.0 | 0.5 | 0.1 |
| Ac | *Acidobacteria; Acidobacteriia* | 20.5 | 46.0 | 64.7 | 44.8 | 1.0 | 4.9 |
| Ac6 | *Acidobacteria*; Subgroup 6 | 0.0 | 0.0 | 0.0 | 0.0 | 0.4 | 0.1 |
| AcAci | *Actinobacteria; Acidimicrobiia* | 2.1 | 2.8 | 1.3 | 0.1 | 0.6 | 0.1 |
| AcAct | *Actinobacteria; Actinobacteria* | 10.6 | 1.8 | 12.1 | 0.4 | 18.4 | 1.0 |
| AcOt | *Actinobacteria*; Others | 0.6 | 20.7 | 4.6 | 0.3 | 0.2 | 0.2 |
| AcTh | *Actinobacteria; Thermoleophilia* | 2.1 | 0.4 | 1.8 | 1.1 | 1.1 | 0.3 |
| Ar | *Armatimonadetes; Armatimonadia* | 0.4 | 0.0 | 0.0 | 0.0 | 0.0 | 0.0 |
| ArC | *Armatimonadetes; Chthonomonadetes* | 0.5 | 0.0 | 0.0 | 0.0 | 0.0 | 0.0 |
| ArFi | *Armatimonadetes; Fimbriimonadia* | 0.6 | 0.0 | 0.0 | 0.0 | 0.1 | 0.0 |
| Bac | *Bacteroidetes; Bacteroidia* | 2.6 | 0.0 | 0.2 | 0.0 | 6.1 | 0.4 |
| Chl | *Chloroflexi; Chloroflexia* | 0.0 | 0.0 | 0.0 | 0.0 | 0.7 | 0.0 |
| CyOt | *Cyanobacteria; Oxyphotobacteria* | 0.0 | 0.0 | 0.0 | 0.0 | 0.3 | 0.0 |
| FirB | *Firmicutes; Bacilli* | 0.0 | 0.0 | 0.3 | 0.0 | 10.7 | 0.7 |
| FirCl | *Firmicutes; Clostridia* | 0.0 | 0.0 | 0.0 | 0.7 | 4.9 | 1.8 |
| Gem | *Gemmatimonadetes; Gemmatimonadetes* | 0.0 | 0.0 | 0.0 | 0.0 | 0.6 | 0.0 |
| GLon | *Gemmatimonadetes; Longimicrobia* | 0.0 | 0.0 | 0.0 | 0.0 | 0.2 | 0.0 |
| Ot | Others; Others | 13.3 | 17.5 | 2.1 | 0.6 | 1.9 | 0.4 |
| PatS | *Patescibacteria; Saccharimonadia* | 0.0 | 0.6 | 1.3 | 0.0 | 0.1 | 0.0 |
| PlPh | *Planctomycetes; Phycisphaerae* | 3.9 | 0.3 | 0.0 | 0.0 | 0.2 | 0.0 |
| PlPl | *Planctomycetes; Planctomycetacia* | 5.7 | 0.0 | 0.7 | 0.0 | 0.7 | 0.0 |
| ProA | *Proteobacteria; Alphaproteobacteria* | 9.4 | 6.3 | 9.7 | 50.2 | 24.6 | 1.7 |
| ProD | *Proteobacteria; Deltaproteobacteria* | 2.9 | 0.6 | 0.0 | 0.0 | 1.6 | 0.1 |
| ProG | *Proteobacteria; Gammaproteobacteria* | 12.7 | 1.6 | 1.1 | 1.8 | 17.7 | 87.5 |
| ProOt | *Proteobacteria*; Others | 1.2 | 0.0 | 0.0 | 0.0 | 0.4 | 0.0 |
| Ver | *Verrucomicrobia; Verrucomicrobiae* | 10.3 | 1.0 | 0.1 | 0.0 | 5.3 | 0.3 |

The descriptions of microbial variety at the order, family and genus level are provided in Figure 5 and Tables S3–S5 and briefly summarized below. However, these levels were not used for correlations with peat chemical parameters. The order *Acidobacteriales* predominated among other orders and constituted 29.6% of all varieties. The orders *Clostridiales* and *Rhizobiales* were the most diverse among other orders and included three families; however, they were present in minor numbers. The order *Pseudomonadales* included two families. The other orders were represented by one family. *Acidobacteriaceae* (*Subgroup 1*) predominated and increased in number with depth, up to 50–60 cm, where the share reached 64.7% and then started to decrease. The next in the number of families was *Burkholderiaceae* (12.6%),

which exhibited a maximum at the peat bottom and mineral horizons (100–110 cm), reaching 87.5% abundance, whereas at the depth of 30–80 it was represented in minor quantities. Families *Sphingomonadaceae*, *Bacillaceae* and *Propionibacteriaceae* were concentrated at the bottom of the frozen peat, in the 90–100 cm layer. *Acetobacteraceae* was evenly distributed over first 60 cm of depth. The families *Chitinophagoceae*, *Acidothermaceae* and *Xanthobacteraceae* were mostly concentrated within the active layer, at the 10–20 cm depth level. The family *Clostridiaceae 1* was mainly present in the mineral layer, at a depth of 100–110 cm. Among all orders, *Clostridiales* and *Rhizobiales* were most diverse and included three families; however, they were present in minor numbers. The order *Pseudomonadales* included two families. The other orders were represented by one family. Microorganisms of the family *Acidobacteriaceae (Subgroup 1)* from the order *Acidobacteriales* constituted the major part of the phylum *Acidobacteria* (17%). The abundance of family members increased in number from the surface to the 50–60 cm layer, where their share reached 58%, and then started to decrease with depth. Among *Proteobacteria*, the maximum abundance was encountered for the family *Burkholderiaceae* (12.6%) from the order *Betaproteobacteriales*. The abundance of microorganisms of this family increased sharply at a depth of 90–110 cm (up to 83.8%), whereas at a depth of 30–80 cm it was present only in minor quantities. The bacteria of families *Sphingomonadaceae*, *Bacillaceae* and *Propionibacteriaceae* were concentrated at a depth of 90 to 100 cm. *Acetobacteraceae* was evenly distributed at a depth of 10–60 cm (first three layers). The remaining families were represented in minor quantities (less 1%) and mainly occurred in the upper layers of the peat core.

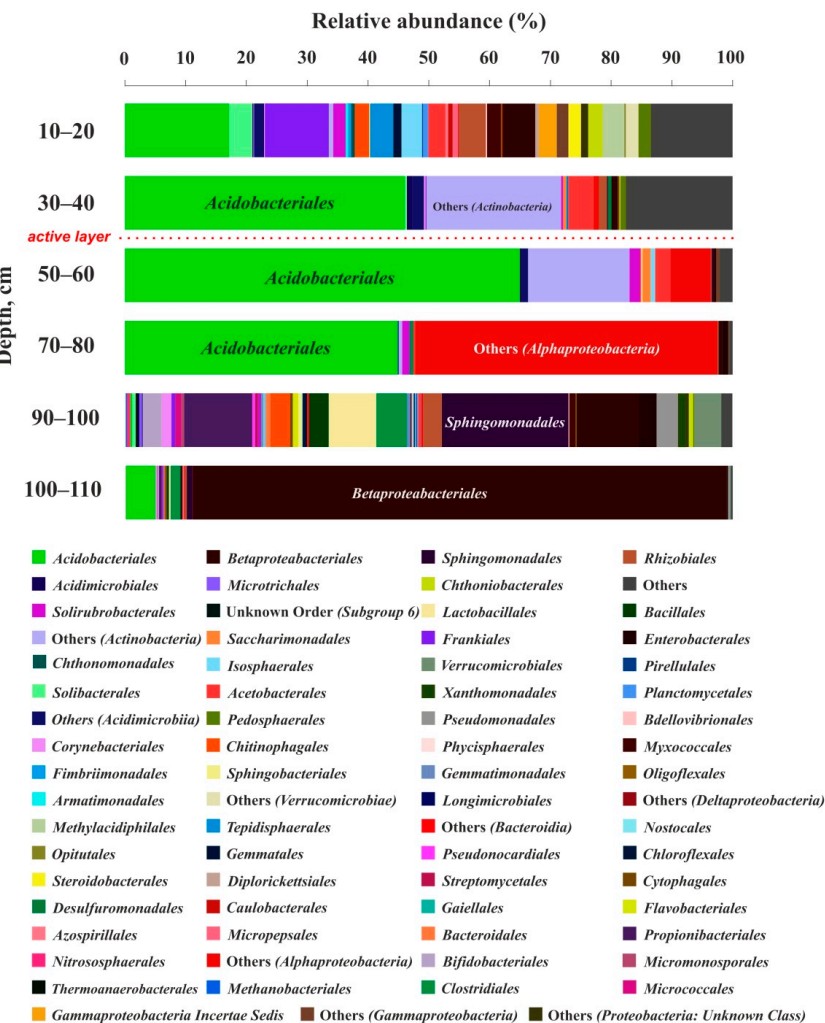

**Figure 5.** Order-level bacterial and archaeal community structure in the peat core based on 16S rRNA.

Generally, the richness of life forms gradually decreased with depth. However, at the bottom of the frozen peat core (90–100 cm layer), it sharply increased. Most genera were represented in minor quantities. Dominant protists belonged to the genera *Granulicella* (13.3%) and *Paucibacter* (10.1%). The genus *Granulicella* was mainly distributed within the 30 to 80 cm depth. The genus *Paucibacter*, on the contrary, was concentrated in the mineral layer, at 100–110 cm depth (87.8%). More than 19% of protists belonged to unknown genera of phyla *Actinobacteria*, *Bacteroidetes*, *Planctomycetes*, *Proteobacteria*, and *Verrucomicrobia*.

### 3.4. Pair Correlation and Multiparametric Statistics of the Full Data Set

First, we checked for linear correlations between specific classes of microorganisms and chemical parameters of peat and peat fluids (water and ice). It can be seen from Table S6 that numerous groups of microorganisms (EuMeth (*Methanobacteria*), ThNit (*Nitrosphaeria*), Chl (*Chloroflexia*), CytOt (*Oxyphotobacteria*), FirB (*Bacilli*), FirCl (*Clostridia*), Gem (*Gemmatimonadetes*)) strongly correlated with macro- (K, Fe, Mg) and micro- (Rb, Mn, Mo, Zn, Ni, Co . . . ) nutrients as well as indifferent geochemical tracers (trivalent and tetravalent hydrolysates) in the peat water and ice. Only a few classes exhibited strong correlations with solid peat chemical composition, such as Fe, Mg, $\delta^{15}$N (AcAci), Ca, Sr (AcOt), macro- and micronutrients, and $\delta^{13}$C (Ar (*Armatimonadia*), ArC (*Chthonomonadetes*), PIPh (*Phycisphaerae*), PIPl (*Planctomycetacia*), ProOt (*Proteobacteria*; other), and Ver (*Verrucomicrobiae*).

In order to achieve further insights into the multiple correlations between physical, chemical and biological parameters, we performed multiparametric statistical treatment of the full dataset, considering all available temperature data as well as peat physical and chemical properties. For this treatment, we considered quantitative data on microorganism numbers and the percentage of different classes in each of the five organic (peat) layers and the one mineral layer. The temperature in various peat layers, the inorganic chemical composition of peat, pore water and ice, the organic (C, N, $\delta^{13}$C, $\delta^{15}$N) composition, and the organic matter quality in each measured layer were also considered. Note that for this treatment, the depth of each layer was considered as an independent variable.

Two PCA treatments of the peat core were performed: with and without the mineral layer. However, due to insufficient data on the mineral layer (lack of ice chemistry and OM quality), the explicatory capacity of PCA for both the organic and mineral part of the core was much lower than that for the peat part solely. As such, we present only the PCA results for the organic part of the peat column.

There were three main factors potentially responsible for the observed variability in the physical, chemical and microbial parameters of the peat core (Table 4, Figure 6). The first factor comprised bottom layers of the column with minimal temperature variations, high peat degradation degree, high contribution of alkyl C, aromatic C and carboxylic C and elevated concentrations of B, Cr and Cu in the peat and DOC, Ca, Mg, K, Si, Fe and various micronutrients (Mn, Ba, Zn, Co, Rb, Mo, B) in the peat water and ice. This factor provided the highest fraction of 16S r-RNA of *Clostridia*, *Bacilli*, *Oxyphotobacteria*, *Gemmatimonadetes*, *Nitrososphaeria*, *Acidobacteria* (Subgroup 6) and *Chloroflexia*. These classes of microorganisms were primarily controlled by peat water and ice chemistry. The second factor acted on the chemical composition of solid peat and marked the minimal proportion of genes of *Verrucomicrobiae*, *Deltaproteobacteria*, *Fimbriimonadia*, *Gammaproteobacteria*, *Phycisphaerae*, *Planctomycetacia*, *Chthonomonadetes*, *Armatimonadia*, *Bacteroidia* and *Acidobacteriia*. These classes avoided the medium part of the peat core, where there was an elevated concentration of lithogenic elements (Ti, Y, REE) and a minimal concentration of some nutrients (K, Rb, V, Mn) and toxicants (As, Pb, Sb). Presumably, these elements marked the presence of silicate dust, and thus the classes of microorganisms of the second factor preferred the peat layers where the proportion of mineral dust was minimal. Finally, the third factor acted on *Actinobacteria*, which were linked to low N and Mo concentration and the highest $\delta^{15}$N and C:N ratio in the peat and low P concentration in water and ice.

**Table 4.** Correlation of factors with peat parameters and share of bacterial taxa (class level).

| Parameter | Abbreviated in Figure 6 | F.1 (52.0%) | Parameter | Abbreviated in Figure 6 | F.2 (27.3%) | Parameter | Abbreviated in Figure 6 | F.3 (13.2%) |
|---|---|---|---|---|---|---|---|---|
| Coordinate a* according to CIE lab (red color) | a* | −0.96 | *Verrucomicrobia; Verrucomicrobiae* | * Ver | −0.99 | N peat | N | −0.86 |
| Depth peat (negative values) | Depth | −0.94 | *Proteobacteria; Deltaproteobacteria* | * ProD | −0.98 | Mo peat | MoP | −0.86 |
| O-alkyl C | COalk | −0.93 | Ti peat | TiP | −0.95 | P ice water | PW | −0.74 |
| SUVA peat ice/water | IW SUVA | −0.88 | *Proteobacteria; Others* | * ProOt | −0.95 | Sr peat | SrP | 0.74 |
| Coordinate b* according to CIE lab (yellow color) | b* | −0.86 | *Armatimonadetes; Fimbriimonadia* | * ArFi | −0.89 | *Actinobacteria; Others* | * AcOt | 0.81 |
| L* CIE lab | L* | −0.85 | K peat | KP | −0.88 | $\delta^{15}$N | 15N | 0.84 |
| Mg peat | MgP | −0.82 | La peat | LaP | −0.86 | C/N | C/N | 0.93 |
| S peat | S | −0.82 | *Proteobacteria; Gammaproteobacteria* | * ProG | −0.85 | | | |
| P peat | PP | −0.77 | Rb peat | RbP | −0.85 | | | |
| Cd peat | CdP | −0.75 | V peat | VP | −0.85 | | | |
| Plab peat | Plab | −0.71 | Cs peat | CsP | −0.85 | | | |
| Maximum temperature in the horizon | MaxAn | −0.70 | *Planctomycetes; Phycisphaerae* | * PlPh | −0.85 | | | |
| Zr peat | ZrP | 0.71 | Mean temperature in the horizon | MeAn | −0.85 | | | |
| Mo ice water | MoW | 0.72 | Ce peat | CeP | −0.84 | | | |
| Nb peat | NbP | 0.76 | Mn peat | MnP | −0.84 | | | |
| Be ice water | BeW | 0.76 | Na peat | NaP | −0.84 | | | |
| U peat | UP | 0.76 | Dy peat | DyP | −0.84 | | | |

**Table 4.** *Cont.*

| Parameter | Abbreviated in Figure 6 | F.1 (52.0%) | Parameter | Abbreviated in Figure 6 | F.2 (27.3%) | Parameter | Abbreviated in Figure 6 | F.3 (13.2%) |
|---|---|---|---|---|---|---|---|---|
| aromatic C | Carom | 0.79 | $\delta^{13}$C peat | 13C | −0.84 | | | |
| B ice water | BW | 0.80 | pH ice water | IW pH | −0.84 | | | |
| Euryarchaeota; Methanobacteria | * EuMeth | 0.81 | Y peat | YP | −0.82 | | | |
| As ice water | AsW | 0.81 | *Planctomycetes; Planctomycetacia* | * PlPh | −0.82 | | | |
| carboxylic C | Ccarb | 0.82 | Pb peat | PbP | −0.82 | | | |
| Dy ice water | DyW | 0.82 | As peat | AsP | −0.81 | | | |
| Ce ice water | CeW | 0.83 | *Armatimonadetes; Chthonomonadetes* | * ArC | −0.80 | | | |

**Table 3**, continued.

| Parameter | Abbreviated in Figure 6 | F.1 (52.0%) | Parameter | Abbreviated in Figure 6 | F.2 (27.3%) | Parameter | Abbreviated in Figure 6 | F.3 (13.2%) |
|---|---|---|---|---|---|---|---|---|
| Y ice water | YW | 0.83 | *Armatimonadetes; Armatimonadia* | * Ar | −0.80 | | | |
| Minimum temperature in the horizon | MinAn | 0.84 | Sb peat | SbP | −0.79 | | | |
| Sb ice water | SbW | 0.84 | Yb peat | YbP | −0.77 | | | |
| Ba peat | BaP | 0.85 | Li peat | LiP | −0.74 | | | |
| Cu ice water | CuW | 0.85 | *Bacteroidetes; Bacteroidia* | * Bac | −0.71 | | | |
| La ice water | LaW | 0.85 | *Acidobacteria; Acidobacteriia* | * Ac | 0.76 | | | |
| Cu peat | CuP | 0.86 | Bulk density peat | BD | 0.79 | | | |
| U ice water | UW | 0.88 | DIC ice water | IW DIC | 0.83 | | | |
| Decomposition of peat | DP | 0.89 | UV$_{254}$ | UV$_{254}$ | 0.84 | | | |
| Alkyl C | Calkyl | 0.90 | | | | | | |

**Table 4.** *Cont.*

| Parameter | Abbreviated in Figure 6 | F.1 (52.0%) | Parameter | Abbreviated in Figure 6 | F.2 (27.3%) | Parameter | Abbreviated in Figure 6 | F.3 (13.2%) |
|---|---|---|---|---|---|---|---|---|
| Nb ice water | NbW | 0.90 | | | | | | |
| *Cyanobacteria; Oxyphotobacteria* | * CyOt | 0.91 | | | | | | |
| *Gemmatimonadetes; Gemmatimonadetes* | * Gem | 0.91 | | | | | | |
| *Thaumarchaeota; Nitrososphaeria* | * ThNit | 0.91 | | | | | | |
| *Acidobacteria*; Subgroup 6 | * Ac6 | 0.91 | | | | | | |
| *Chloroflexi; Chloroflexia* | * Chl | 0.91 | | | | | | |
| Zr ice peat | ZrW | 0.91 | | | | | | |
| Yb ice peat | YbW | 0.91 | | | | | | |
| *Firmicutes; Bacilli* | * FirB | 0.91 | | | | | | |
| Cr ice peat | CrW | 0.92 | | | | | | |
| B peat | BP | 0.92 | | | | | | |
| Cr peat | CrP | 0.92 | | | | | | |
| Ti ice peat | TiW | 0.93 | | | | | | |
| Specific conductivity of peat ice waters | IW EC | 0.93 | | | | | | |
| Mn ice peat | MnW | 0.93 | | | | | | |
| Ba ice peat | BaW | 0.94 | | | | | | |
| *Firmicutes; Clostridia* | * FirCl | 0.94 | | | | | | |

**Table 4.** *Cont.*

| Parameter | Abbreviated in Figure 6 | F.1 (52.0%) |
|---|---|---|
| V ice peat | VW | 0.95 |
| Cs ice peat | CsW | 0.95 |
| Na ice peat | NaW | 0.96 |
| Ca ice peat | CaW | 0.96 |
| Li ice peat | LiW | 0.96 |
| Fe ice peat | FeW | 0.96 |
| Mg ice peat | MgW | 0.97 |
| Cd ice peat | CdW | 0.97 |
| K ice peat | KW | 0.97 |
| Rb ice peat | RbW | 0.98 |
| Zn ice peat | ZnW | 0.98 |
| Co ice peat | CoW | 0.99 |
| Ni ice peat | NiW | 0.99 |
| Sr ice peat | SrW | 0.99 |
| DOC ice peat | IW DOC | 0.99 |
| Si ice peat | SiW | 1.00 |

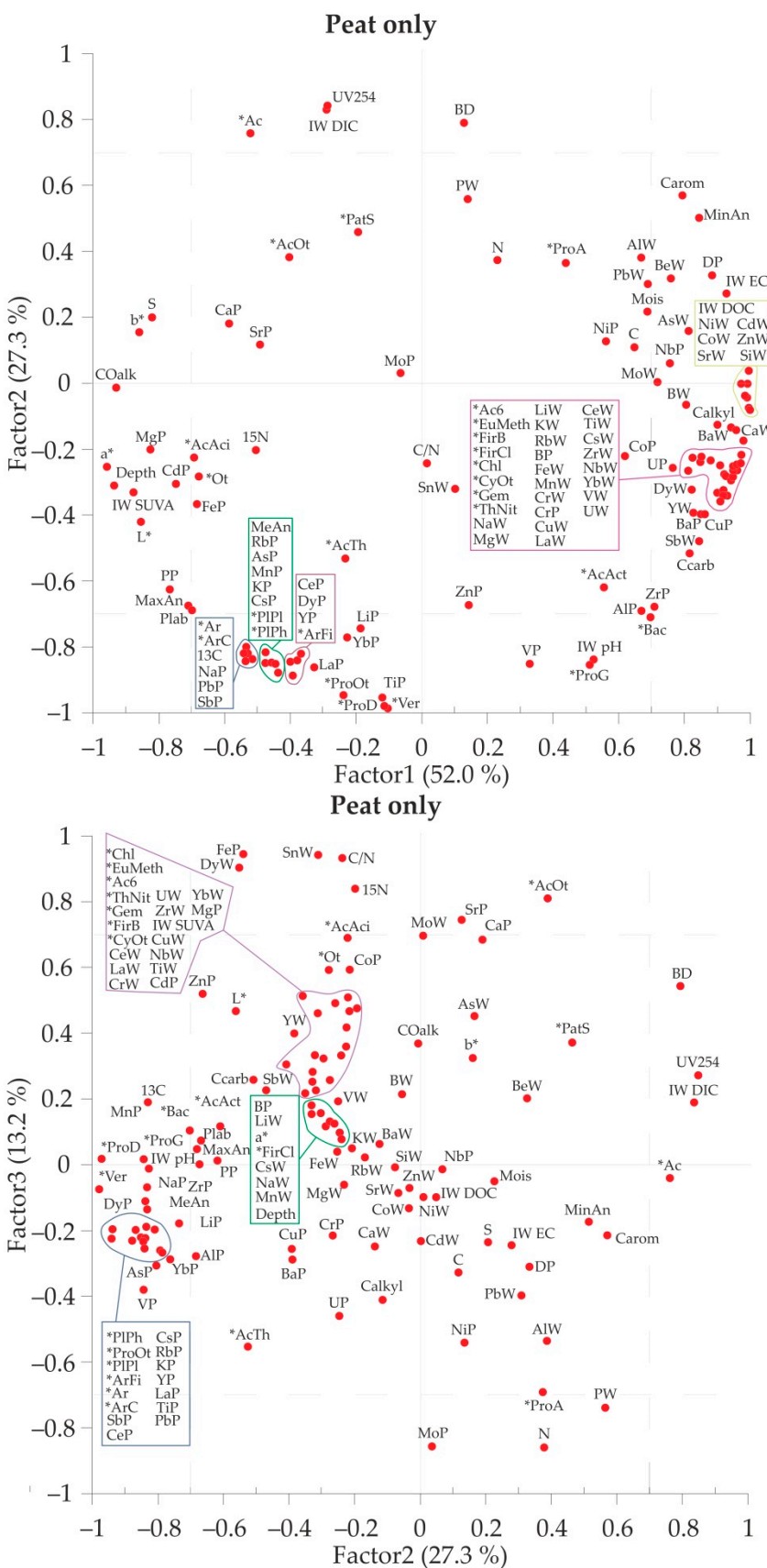

**Figure 6.** Results of PCA on peat (and peat water/ice), without mineral layer. A plot of F1 × F2 and F2 × F3 factors (upper and low panel, respectively). The abbreviations used in the Figure are detailed in Table 3.

## 4. Discussion

### 4.1. Elementary Peat and Peat Fluid Composition and Organic Matter Quality

As also noted in the previous study of the peat core from a neighboring mound [20], the inter-layer variations of most major and trace elements in the frozen and thawed peat of the peat bog mound in this study are within the average elementary peat composition of western Siberia across the more than 1500 km latitudinal transect [55]. In particular, an enrichment of the upper 0–20 cm horizon in macro- (P, K, Mg) and micro-nutrients (Mn, Zn, Rb) as well as Cs and potential toxicants of atmospheric origin (As, Cd, Sb, Tl, Pb) reflects the role of ground vegetation in providing these elements via vertical downward water migration. As such, the sampled peat core may be considered representative of the permafrost-affected part of western Siberia. For most elements, the bottom (frozen) part of the peat core exhibited a composition that was quite similar to the upper (thawed) part of the core. Using a Mann-Whitney test, significant ($p < 0.05$) differences between frozen and thawed layers were observed only for C:N ratio, Na, Mg, K, Al, Ca, Zn, Ge, Hf, W, Nb, Pb, U, Ba, Cd, Sb and Th. The differences ($p < 0.05$) in element concentration between peat ice and peat pore water were observed for S.C., SUVA, Li, Al, Si, P, Fe, Co, Rb, Cs, DOC, Ca, Sr, Mg, Ni, Cd and Tl.

Concerning peat pore water and ice chemical composition, a local maximum of DOC, UV absorbency, specific conductivity and some trace element concentrations at 40–50 cm depth (Figure 3) suggested some solute and organic colloid concentrations via freezing front migration, as demonstrated in the peat ice study across the WSL [58]. However, this pattern of concentration was different for other nutrients, such as phosphorus, which exhibited a peak at 80 cm depth. Finally, the downward increase in the concentration of micro-nutrients (B, K, Mn, Ni, Zn, Mo, Ba) suggested their possible use for biota in case of a massive ice peat thaw. In contrast, enrichment of peat pore waters from the surface layer (0–10 cm) in toxicants (As, Sb, Pb) suggested deposition of these elements from the atmosphere.

The increase of alkyl C:O alkyl C ratio and of aromatic C proportion with depth along the peat profile may reflect an advanced degree of OM degradation [61]. From 0 to 35 cm depth, lichen and sphagnum are the main contributors, whereas, from 65 to 95 cm depth, wood pieces and shrub dominate the plant remains in the peat [20]. As a result, an increase in aromatic C (in relation to phenols from lignin) and a decrease in O-alkyl C proportions may also result from the variable contribution of different plant species to peat formation [62,63]. Except for the 19–21 cm depth, which presents slightly higher value, the $\delta^{15}N$ along the peat core is in the range given by Amundson et al. [64] for the soils of this climatic region, and the absence of isotopic enrichment with depth is also consistent with data from water-saturated soils [65].

### 4.2. Bacterial Number and Genetic Diversity of Soil Microorganisms

Numerous data available for the bacterial counts in Siberian soils demonstrated that the total number of cells generally decreases with the depth ([66] and references therein). In accordance with recent studies of enzyme patterns and microbial community compositions in western Siberian soils [34], as well as metabolic activity with respect to various organic substrates assessed by the Biolog Ecoplate technique [20], we demonstrated that each peat soil horizon harbored different microbial consortia.

The total number of cells of microorganisms quantified in this study (from $1.35 \times 10^{10}$ at the surface to $2.27 \times 10^7$ cell $g_{peat}^{-1}$ in the mineral layer) is comparable to the total viable bacterial count of a similar peat core sampled three years before, as measured by the DAPI technique ($73 - 1.6) \times 10^7$ cell $g_{peat}^{-1}$ [20]. The latter values are within the range measured in the Antarctic and High Canadian Arctic permafrost ($10^5$–$10^7$ cells/$g_{soil}$ [48,67] and $(2-4) \times 10^7$ cell/$g_{soil}$ [1]) and Siberian permafrost ($10^7$–$10^8$ cells/$g_{soil}$ [68,69]). Total cell count ranged from $9.32 \times 10^6$ cells per g dry weight to $1.37 \times 10^7$ cell/$g_{soil}$ in the permafrost samples from Alaska [70]. Note, however, the quite high number of microbial

cells in the surface layer of peat core, which exceeds the typical values of mineral soils by a factor of 100 to 1000.

In agreement with Jansson and Taş [69], the diversity of bacteria in the permafrost peat assessed in this study is much higher than that of archaea (99% of all sequences in this study). The dominant phylum *Proteobacteria*, *Firmicutes*, *Acidobacteria* and *Actinobacteria* identified in the present study are also reported in other permafrost environments [36,69]. The microbial population of the peat soil of Central Siberia was dominated by bacteria (96.6–98.7% of all sequences) and included *Proteobacteria* (34–25%), *Actinobacteria* (20–18%), *Acidobacteria* (8–14%) and *Verrucomicrobia* (13–14%), as reported by Grodnitskaya et al. [33]. Similarly, polygonal peat bogs of the Canadian Arctic contained *Proteobacteria* (40–50%), *Bacteroidetes* (20–40%) and *Acidobacteria* (10–15%) [71].

In studied peat core, *Proteobacteria* were abundant in all layers and accounted for 47% of all sequences, on average. This was represented by *Alphaproteobacteria* (5–50%) and *Gammaproteobacteria*, with maximal abundance at the mineral layer (84%). The possibility of *Proteobacteria* synthesizing DNA at a temperature from 0 to $-20$ °C [72] may explain their abundance at the 60–110 cm depth, where the temperature ranged from $-0.1$ to $-0.5$ °C. It is noteworthy that *Alphaproteobacteria* are capable of growing after nitrogen addition [35], which can explain the dominance of this class at 70–80 cm depth, where the N concentration achieves its maximum. The main representative of *Alphaproteobacteria* is clade *Sphingomonades*, which is abundant at the depth of 90 to 100 cm (20.3%). This clade is known to degrade lignocellulose debris [73], which is consistent with the botanical composition of studied peat: the wood debris is highly abundant below 80 cm depth. The representatives of clade *Rhizobiales* are highly abundant at 0–20 cm depth. This order includes gram-negative soil bacteria, capable of fixing nitrogen in symbiosis with plants [74–79]. Overall, the near-surface layer of peat is the major location for root exudates, most likely in the form of labile C substances, which are known to favor the growth of copiotrophic microorganisms [23] such as *Gamma-* and *Alphaproteobacteria*, as also evidenced from a study of catabolic activity of peat microorganisms [20].

Class *Acidobacteria* is highly abundant in soils and may constitute up to 52% of the microbial community [80]. In studied peat core, oligotrophic *Acidobacteria* were highly abundant and mostly present in the thawed layer or permafrost boundary (50–60 cm depth) and strongly decreased in the deep-frozen and mineral horizon. This may reflect the need of these bacteria for nutrients, which are concentrating at this boundary. Moreover, the peat water from 50–60 cm depth exhibited the lowest pH of all sampled horizons, consistent with the preference of *Acidobacteria* for acidic environments.

The *Actinobacteria* were present over the full depth of the peat core, which might be due to their ability to assimilate a broad spectrum of organic substrates and their adaptation to low temperature, for example, due to elevated content of guanidine and cytosine [80,81]. The *Bacteroidetes* are the main representatives of gram-negative anaerobic bacteria. Their localization at 90–100 cm depth reflects both the anaerobic conditions at the bottom of the peat core and the fact that this phylum includes several psychrophilic bacteria. Note that *Bacteroidetes* is one of the dominant phyla in polygonal tundra soils of the Lena Delta [71]. Finally, the deep 90–100 layer of frozen peat also harbored other anaerobic spore-forming bacteria, such as *Clostridia* (4.9%) and *Bacilli* (10.7%).

### 4.3. Multiple Factors Governing the Microbial Diversity in the Peat Core

The chemical composition of peat porewaters and suprapermafrost waters sampled in different sites of the Khanymey region (i.e., mounds, depressions) was shown to be rather similar, and overall in western Siberia, the impact of latitude and permafrost was much stronger than the difference between micro-landscapes (Raudina et al., 2018). Because the fluid phase of the peat layer represents maximal possible variations in chemical composition among soil micro-environments, we believe that the difference in the chemical and microbial composition between adjacent cores is quite small.

Overall, in our approach to microbial diversity, we followed the methodology of a previous study of the peat core from the Khanymey site of the WSL (Morgalev et al., 2017). The peat on the watershed divide of the Khanymey Research Station is highly homogeneous.

However, we admit that a special study of microbial diversity in peat cores sampled in different micro-landscapes and across the permafrost gradient is needed to extend the obtained results to a much larger territory of western Siberia.

The decrease in the bacterial number (this study) and AWCD [20] with depth, primarily within the active layer, probably occurred due to a decrease in the number of consumed substrates. In the 0–20 cm upper layer of peat, which contained lichen debris, the most metabolically active bacteria were those that consumed amines such as *Proteobacteria* [82]. In a previous study of metabolic activity of peat microorganisms, we demonstrated a peak of activity at the border of the thawed layer and permafrost [20]. It is not excluded that the maximal variety of *Acidobacteria*, also occurring at this layer (50–60 cm) in this study, corresponds to the maximal capacity of these oligotrophic bacteria to process organic substrates. It could not be proven, however, that the peak of metabolic activity and the highest diversity of *Acidobacteria* at the permafrost–thaw layer boundary are linked to the specificity of organic carbon quality in this layer or the chemical composition of peat pore water, since no direct correlation between these parameters has been established.

An increase in Ni concentration in peat ice below 70–80 cm depth might be consistent with its involvement in the metabolism of anaerobic methane-producing bacteria, which become abundant at the bottom of the peat core. The deep layers of strongly degraded peat contain elevated B, Cr, Cu and Ba concentrations, aromatic C, carboxylic C and alkyl C and are also enriched in DOC, Ca, Mg, K, Fe, Si, Zn, Co, Ni, Mn, V and Rb in the liquid (interstitial water and ice) phase. The latter favored the elevated proportions of bacterial classes *Clostridia*, *Bacilli*, *Oxyphotobacteria*, *Gemmatimonadetes*, *Chloroflexia* and Archae *Thaumarchaeota*, and *Nitrososphaeria*. Several classes of bacteria demonstrated a tendency to avoid the medium part of the peat profile, located just below the active layer boundary, where there were minimal concentrations of lithogenic elements linked to mineral silicate dust. These were *Verrucomicrobiae*, *Deltaproteobacteria*, *Fimbriimonadia*, *Gammaproteobacteria*, *Phycisphaerae*, *Planctomycetacia*, *Chthonomonadetes*, *Armatimonadia*, *Bacteroidia* and *Acidobacteriia*. Finally, a single group of *Actinobacteria* was associated with minimal N concentration and the highest $\delta^{15}$N. Presumably, it actively consumed N-rich compounds for guanine and cytosine synthesis, which is consistent with a metabolic diversity of the studied peat column [20]. It is also possible that, after cell lysis, the bacterial N is removed and translocated from the relevant peat layer.

Placing the obtained results in the context of permafrost thaw and active layer deepening, we hypothesize that freezing front propagation downwards leads to mobilization of essential macro- and micro-nutrients at the active layer boundary; as a result, the layer of maximal genetic diversity of the microbial population will move down the soil profile. Moreover, the total microbial number strongly decreased ($\times$x 10) from the surface to the active layer and further to the mineral horizon ($\times$ 10). We, therefore, estimate that, under a climate warming and permafrost thaw scenario, in the case of full thawing of the frozen peat and underlying mineral horizons, the total number of bacteria over the 100 cm thick layer may significantly increase.

## 5. Conclusions

Unlike the total bacterial number, which decreased almost two orders of magnitude from the surface down to mineral frozen layer of the peat core, the genetic diversity of microorganisms increased with depth. Moreover, consistent with previously reported peaks of the intensity of substrate utilization right at the boundary of the active layer and frozen peat, we evidenced a dominance of *Acidobacteria* in the genetic diversity of microorganisms of the peat core at the thawed peat–permafrost boundary.

A multi-parametric statistical analysis demonstrated that three factors were responsible for the main variability of chemical and microbial parameter distribution. There was a strongly pronounced preference of *Clostridia*, *Bacilli*, *Oxyphotobacteria*, *Gemmatimonadetes*, *Thaumarchaeota*, *Nitrososphaeria* and *Chloroflexia* for deeper peat horizons, which were enriched in alkyl C, carboxyl C, aromatic C, B, Cr, and Cu in the solid phase and DOC and various macro- and micro-nutrients in the fluid (peat porewater and ice) phase. In contrast, classes *Verrucomicrobiae*, *Deltaproteobacteria*, *Fimbriimonadia*, *Gammaproteobacteria*, *Physisphaerae*, *Planctomycetacia*, *Chthonomonadetes*, *Armatimonadia*, *Bacteroidia* and *Acidobacteriia* preferentially avoided the intermediate part of the peat core, where there was a minimal concentration of mineral dust. *Actinobacteria* demonstrated a strong link with the highest C:N ratio and $\delta^{15}$N value.

Given the largest diversity of microorganisms in the bottom organic (80–100 cm) and mineral (100–110 cm) layers, the frozen peat and underlying mineral substrates of western Siberia present a high potential for microbial diversity; in addition to its sizeable nutrient (mineral and organic) resources, this area may represent an important source of active microorganisms under continuous permafrost thaw.

**Supplementary Materials:** The following are available online at https://www.mdpi.com/article/10.3390/d13070328/s1, Table S1: Chemical composition of peat interstitial water and ice; Table S2: Number of microorganisms as a function of peat depth; Table S3: Relative proportion of order units of peat microorganisms over the peat column; Table S4: Family units of peat microorganisms; Table S5: Genus units of peat microorganisms; Table S6: Most pronounced pairwise correlations between class of microorganisms and peat and peat water/ice chemical components.

**Author Contributions:** Conceptualization, O.S.P., A.S.A. and L.S.S.; writing—original draft preparation, O.S.P.; sample collection, A.G.L., D.K. and S.V.L.; organic matter quality and isotopic composition, S.P., M.A.A., M.C.-R. and S.V.L.; molecular genetic analysis, O.Y.K.; writing—review and editing (microbiology part), S.N.K.; visualization and supervision (genetic part), A.S.A.; chemical analyses, A.G.L., S.V.L., D.K. and L.S.S. All authors have read and agreed to the published version of the manuscript.

**Funding:** The research was funded by the Government of the Russian Federation in compliance with the Resolution of 09 April 2010 No. 220 (the contract from 14.03.2017 No. 14.Y26.31.0009). The authors declare that they have no financial or personal relationships that may have inappropriately influenced them in writing this article.

**Institutional Review Board Statement:** Not applicable.

**Informed Consent Statement:** Not applicable.

**Data Availability Statement:** The data presented in this study are available on request from the corresponding author.

**Acknowledgments:** The research was done using equipment of the Core Centrum 'Genomic Technologies, Proteomics and Cell Biology' in ARRIAM.

**Conflicts of Interest:** The authors declare that there is no conflict of interest related to the publication of this article, which should be reported.

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
