# Peer review of "Bacterial Number and Genetic Diversity in a Permafrost Peatland (Western Siberia): Testing a Link with Organic Matter Quality and Elementary Composition of a Peat Soil Profile"

_diversity, doi:10.3390/d13070328_

Round 1

Reviewer 1 Report

Dear Authors,

the investigation of “Bacterial number and genetic diversity in a permafrost peatland (western Siberia)” is a very current topic considering the news of global warming. The research is very interesting and well presented. I found that manuscript is well written, easy to read and complete in every section. Below were suggested only a few corrections that I hope could improve the manuscript.

-Lane 194: capital letter for V3-V4

-Lane 241: make uniform ppb for all chemical elements in all columns in Table S1

-Lane 249: is no clear reference to Figure 2

-Lane 364: delete between

-Lane 367: phyla

-Lane 401: 16S with a capital letter

-Lane 434: delete space

-Lane 514: layer 90-100 in Table S4

-Lane 518: Clostridia (4.9%) and Bacilli (10.7%) in Table S4

Best regards

Author Response

Dear reviewer!

Thank you very much for your comments and suggestions to our work. It helped us to improve the quality of our article. We are most grateful to you for helping us. Please find attached a revised version of the manuscript diversity-1269009.

Line 194: capital letter for V3-V4

A correction was made, thank you

Line 241: make uniform ppb for all chemical elements in all columns in Table S1

A correction in Table S1 was made. The table has been transferred to the manuscript

Line 249: is no clear reference to Figure 2

We made a mistake when adding a reference. We replaced it with the correct one - Table 1. Thank you for this important remark

Line 364: delete between

A correction was made, thank you

Line 367: phyla

A correction was made, thank you

Line 401: 16S with a capital letter

A correction was made, thank you

Line 434: delete space

A correction was made, thank you

Line 514: layer 90-100 in Table S4

A correction was made, thank you

Line 518: Clostridia (4.9%) and Bacilli (10.7%) in Table S4

A correction was made, thank you

Reviewer 2 Report

This article by Aksenov et al. deals with the characterization of the both organic matter and bacterial communities in a permafrost peatland located in western Siberia.  The theme of this study fits correctly with the scopes of Diversity. Given the amount of C stored in permafrost and their potential feedback to global change, it is essential to improve our knowledge on the microbial communities living in those environments.  Despite the high number of analyses done, I am afraid that this manuscript cannot be considered for publication. The main reason for this is the absence of replicates. Although it is stated that this work is a preliminary study, no conclusion can be drawn on the results obtained on only one sample. Even though the authors argue that the “selected peat core is highly representative for the region”, this assertion should be verified. I suggest the authors to clarify the scientific question and use this very interesting site to investigate deeper the relationships between OM and microbial communities, using less analytical methods but more replicates (at least 3 composite samples).

Author Response

Dear reviewer!

Let us thank you very much for the professional comments and suggestions to our work. Please find below our point-by-point itemized answers and corrections.

Despite the high number of analyses done, I am afraid that this manuscript cannot be considered for publication. The main reason for this is the absence of replicates. Although it is stated that this work is a preliminary study, no conclusion can be drawn on the results obtained on only one sample. Even though the authors argue that the “selected peat core is highly representative for the region”, this assertion should be verified. I suggest the authors to clarify the scientific question and use this very interesting site to investigate deeper the relationships between OM and microbial communities, using less analytical methods but more replicates (at least 3 composite samples).

We thank the reviewer for this important point.

The reviewer correctly pointed out that composite samples are needed to assess the chemical control on microbial communities. We would like to note that our 5 to 10 cm layers of 25 to 50 cm3 volume correspond to quite composite samples, correctly representing the peat structure and composition.

As for the representability of studied peat core, questioned by the reviewer, our arguments are based on similarity of chemical composition of both solid and fluid phase of the cores sampled in this region. The peat on the watershed divide of the Khanymey Research Station is highly homogeneous in space, and sampling several adjacent cores of the same study site would not provide any new insights on lateral chemical and microbial diversity.

The chemical composition of peat porewaters and suprapermafrost waters sampled in different sites of the Khanymey region (i.e., mounds, depressions) was shown to be rather similar, and overall in western Siberia, the impact of latitude and permafrost was much stronger than the difference between micro-landscapes (Raudina et al., 2017, 2018). Because the fluid phase of the peat layer represents maximal possible variations in chemical composition among soil micro-environments, we believe that the difference in chemical and microbial composition between adjacent cores would be quite small.

Overall, in our approach for microbial diversity, we followed the methodlogy of previous study of the peat core from the Khanymey site of the WSL (Morgalev et al., 2017). The peat on the watershed divide of the Khanymey Research Station is highly homogeneous.

To further strengthen our response, we have prepared an appendix based on published and unpublished materials confirming the similarity of the composition of the cores in the study area at a distance of more than 2 km (see Appendix).

In addition, it should be noted that the flat-mound bogs examined in different parts of the zone, although somewhat different in their structure, exhibit many similar features. Namely, sedge, sedge-hypnum, sphagnum and hypnum types of peat with a large admixture in the upper layer of heather shrubs, and in the underlying layers - woody remains, horsetail, in some places Menyanthes, Scheuchzeria, Eriophorum, prevail throughout the peat deposit. The degree of decomposition throughout the entire thickness of the deposit varies within 5–30%, in the bottom layer in some places it reaches 35–50%. The homogeneous spatial structure of the peat deposit of autonomous interfluvial deposits of hillocks of flat-hilly massifs is also associated with the peculiarities of the upper part of the sandy sediments underlying the frozen bogs. These deposits have spatially monotonic properties, since they were overwhelmed in the Late Pleistocene (Velichko, 2011, http://dx.doi.org/10.1016/j.quaint.2011.01.013). Due to this, the peat bogs formed on these deposits also had a similar initial trophicity regime, which caused a similar botanical composition in large areas.

However, we admit that a special study of microbial diversity in peat cores sampled in different micro-landscapes and across the permafrost gradient is needed to extend the obtained results to much larger territory of western Siberia.

Reviewer 3 Report

The authors describe the microbial population of various layers of a peat core and its interaction with the organic and inorganic composition of both solid and liquid compartments of the core sampled. The work is very interesting, rich of analyses and interesting results.

The introduction is well written, hypotheses and aims of the study are clearly pointed out.

Materials and methods are described in detail, however no clear information are reported on the number of samples collected or the number of replicates of soil or microbiological analyses carried out.

All the abbreviations reported in the text, as  SUVA254, (Specific UV absorptivity at 254 nm), AWCD (Average well color density), DOC, DIC … have to be reported in full, the first time they are mentioned in the text.

The main criticism concerns the great effort that reading this work requires.

This depends on the authors' choices of graphing the results, and of reporting the data shortening the names of taxa. I suggest to avoid such abbreviations. The main recommendation to the authors is to find a solution to make easier the reading of the manus and the data representation. Figure 5, as an example, is not readable, as it is impossible to distinguish among so many colors.

Too much details that are too tiring to read are reported at lines 331-368, regarding supplementary data. Supplementary data generally do not require a so detailed description in the results paragraph. If you consider these data so important to be so detailed reported, they should have not to be reported as supplementary data.

Figures 6 and 7 – All the abbreviations reported in these figures would have to be reported in the captions, in order to facilitate the reader

Some minor changes:

Line 249 – Fig. 2 shows only carbon and nitrogen values while macro- and micronutrients values are not showed

Lines 245-249 – could you please verify the actual correspondence between some values and those reported in Table S1?

Lines 258-259 – Perhaps it is better to add the reference to table S2

Line 294- replace (100-100 cm) with (100-110 cm)

Line 334 – Where is the “17.2%” value reported?

Line 489 – Do you mean DNA?

Line 512 – I did not find in Kielak et al. (2016) references to guanidine and cytosine contents. Are you sure about this reference?

Line 560- Mineral instead of minela

561 – microorganisms instead of microorganism

Author Response

Dear reviewer!

Thank you very much for the professional comments and suggestions to our work. Taking into account your comments, we have raised the level of our article. Please find below our point-by-point itemized answers and corrections.

Line 249 – Fig. 2 shows only carbon and nitrogen values while macro- and micronutrients values are not showed

We thank the reviewer for this important point. We made a mistake when adding a reference. Correct reference is to the Table 1. We made a correction.

Lines 245-249 – could you please verify the actual correspondence between some values and those reported in Table S1?

A correction was made, thank you

Lines 258-259 – Perhaps it is better to add the reference to table S2

We thank the reviewer for this important addition. We added a reference to the Table S2

Line 294- replace (100-100 cm) with (100-110 cm)

A correction was made, thank you

Line 334 – Where is the “17.2%” value reported?

We thank the reviewer very much for this valuable comment. There was made a mistake, that data concerned the total value. We corrected this point and added the actual value

Line 489 – Do you mean DNA?

A correction was made, thank you

Line 512 – I did not find in Kielak et al. (2016) references to guanidine and cytosine contents. Are you sure about this reference?

Article by Kielak et al. (2016) mainly relates to "to assimilate a broad spectrum of organic substrates". For the point "elevated content of guanidine and cytosine" mentioned by the reviewer, we added the following article: Ventura et al. (2007), which seems to have been missed earlier. Thanks you for the help.

Line 560- Mineral instead of minela

A correction was made, thank you

Line 561 – microorganisms instead of microorganism

A correction was made, thank you

Materials and methods are described in detail, however no clear information are reported on the number of samples collected or the number of replicates of soil or microbiological analyses carried out.

We thank the reviewer for this important point. We examined one peat core. Chemical analyzes were carried out in several replicates. In section 2 Study Site and Methods, in clause 2.5 Statistical treatment, we explained such a decision and provided a reference to a source supporting our position.

Cite from clause 2.5 "Note that selected peat core is highly representative for the region of WSL permafrost peatlands. Multiple cores sampled within the same test site of Khanymey (discontinuous permafrost zone), from different mounds, demonstrated high similarity of chemical composition of both pore water and peat ice [58]. Due to high homogeneity of peat and underlaying mineral substrate of the region, this suggests a possibility of using a single soil core for representing chemical diversity of solid and liquid phase and metabolic activity of microorganisms (i.e. [20])."

All the abbreviations reported in the text, as  SUVA254, (Specific UV absorptivity at 254 nm), AWCD (Average well color density), DOC, DIC … have to be reported in full, the first time they are mentioned in the text.

We thank the reviewer for this important remark. We added transcripts for abbreviations when they were first mentioned.

The main recommendation to the authors is to find a solution to make easier the reading of the manuscript and the data representation. Figure 5, as an example, is not readable, as it is impossible to distinguish among so many colors

We understand this concern of the reviewer and we agree that Fig. 5 is a bit difficult to read. However, this is a standard and most commonly used way of presenting the biodiversity of microorganisms. To facilitate the reading of this figure, we added the names of dominant orders in each layer directly to the histograms. We also improved the presentation of Fig. 4.

Too much details that are too tiring to read are reported at lines 331-368, regarding supplementary data. Supplementary data generally do not require a so detailed description in the results paragraph. If you consider these data so important to be so detailed reported, they should have not to be reported as supplementary data.

We thank the reviewer for pointing out this inconsistency. We shifted a big deal of information from the Supplement to the main text to facilitate the reading of the manuscript. We do need to present and properly describe these data in the main text because they i) characterize the physico-chemical properties of the peat core investigated in this study and ii) are used for the most essential and novel part of this work - relating the diversity of microorganisms to chemistry of the peat and peat porewaters. As such all the data described in the text are used for multi-parametric statistics (PCA) which allow providing new information on environmental control of the biodiversity in permafrost peatlands.

Figures 6 and 7 – All the abbreviations reported in these figures would have to be reported in the captions, in order to facilitate the reader

Good point. All the abbreviations used in Figure 6a and 6b (there is no Fig. 7) are listed in Table 3, and we added a relevant sentence to the figure caption.

Round 2

Reviewer 2 Report

I would like to thank the co-authors for their answer. Unfortunately, I'm afraid that explanation isn't enough.Ecological sciences are based on the observation of variability, whether spatial or temporal, structural or functional, of organisms in their environment. I cannot support the publication of results without replicates. If the authors are not able to provide additional data to confirm the observed trends, I suggest rejecting this manuscript.

Author Response

Dear reviewer!

Let us thank you very much for the professional comments.

Our work is designed in the field of soil science where the replicates are considered in vertical space. The patterns of soil chemistry and microbiology are highly stable in this vertical space and can be considered as true fingerprints of the soil. A lot of significant and breakthrough results in soil science were obtained based on a single soil profile, provided that thorough and multidisciplinary analyses are available.

We did confirm the repeatability of the chemistry of the soil profile, both for the solid and fluid phase (see our responses to the 1st round of review). Performing full genetic analysis of all horizons in vertical replicates will double or triple the amount of work which is already voluminous: we would like to point out that our study reports an unprecedented combination of water and soil chemical composition of major and trace elements, state of the art organic matter characterization including  NMR analyses, and full genetic analysis of multiple horizons. As requested by the reviewer, we did provide additional data which confirm the vertical pattern of soil chemistry. What is more important, the lateral heterogeneity of peat in permafrost peatlands of western Siberia is really low as we confirm in numerous publications of our group.

Reviewer 3 Report

Dear authors, 

thank you for your answer and changes to the manus. I have no additional recommendations and I endorse the manus publication.

Be careful in the draft to correct the typing error on line 544 'methodlogy'

Author Response

Dear reviewer!

We would like to thank you again for your help. Due to you, we have increased the value and reliability of our manuscript, as well as been able to make it more accessible for readers to understand. Your contributions to our manuscript have been incredibly valuable and helpful.

Correction on the line 544 in the word "methodology" is made, thank you for your attention!